# Allosteric nanobodies to study the interactions between SOS1 and RAS

Baptiste Fischer [1,2], Tomasz Uchański[3,4], Aidana Sheryazdanova [5,6], Simon Gonzalez[7], Alexander N. Volkov [3,8], Elke Brosens[3,4], Thomas Zögg [3,4], Valentina Kalichuk [3,4], Steven Ballet [7], Wim Versées [3,4], Anna A. Sablina[5,6], Els Pardon [3,4], Alexandre Wohlkönig[3,4] & Jan Steyaert [3,4] ✉

Protein-protein interactions (PPIs) are central in cell metabolism but research tools for the structural and functional characterization of these PPIs are often missing. Here we introduce broadly applicable immunization (Cross-link PPIs and immunize llamas, ChILL) and selection strategies (Display and co-selection, DisCO) for the discovery of diverse nanobodies that either stabilize or disrupt PPIs in a single experiment. We apply ChILL and DisCO to identify competitive, connective, or fully allosteric nanobodies that inhibit or facilitate the formation of the SOS1•RAS complex and modulate the nucleotide exchange rate on this pivotal GTPase in vitro as well as RAS signalling *in cellulo*. One of these connective nanobodies fills a cavity that was previously identified as the binding pocket for a series of therapeutic lead compounds. The long complementarity-determining region (CDR3) that penetrates this binding pocket serves as pharmacophore for extending the repertoire of potential leads.

Although protein-protein interactions (PPIs) are considered as attractive pharmaceutical targets, they remain largely unexploited[1]. Several methods have been developed for studying PPIs in vitro and in vivo, such as complementation, NanoBiT, NanoBRET, FRET, AlphaLISA, ELISA, ITC, and SPR, but adequate tools are missing to modulate these interactions. Here, we introduce a simple but robust approach for the simultaneous discovery of competitive, connective, and allosteric nanobodies that modulate, disrupt, or stabilize transient PPIs. Nanobodies (Nbs) are the small (15 kDa) and stable single-domain fragments harboring the full antigen-binding capacity of the original heavy chain-only antibodies that naturally occur in Camelids. Nbs are increasingly used as research tools in biotechnology and applied in diagnosis and therapy[2]. Because they access conformational epitopes in diverse cavities and hinge regions, Nbs are also increasingly utilized in structural biology to freeze dynamic proteins into single functional conformations[3].

Pivotal for our approach, any association of two or more interacting proteins generates novel structural features that are unique to the complex. New composite surfaces are formed, and allosteric sites may undergo conformational changes upon association, disclosing altered conformational epitopes that are not exposed on the separate protomers. According to this principle, many PPIs are regulated by competitive, allosteric or connective modulators that bind these differentiating structural features, with the main mechanistic classification being a division into PPI inhibitors and stabilizers[4] (Supplementary Fig. 1a). Thermodynamic cycles offer a simple framework for analyzing the possible effects of antibodies that bind components of such PPIs (Supplementary Fig. 1b). Any antibody that selectively binds an allosteric or connective epitope that is unique to the complex will stabilize the PPI proportionally to its affinity for the complex relative to its affinity for the protomers. Inversely, antibodies that selectively bind to

[1]Université de Bordeaux, CNRS, Bordeaux INP, CBMN, UMR 5248, Pessac, France. [2]European Institute of Chemistry and Biology (IECB), 2 rue Robert Escarpit, Pessac, France. [3]VIB-VUB Center for Structural Biology, VIB, Pleinlaan 2, Brussels, Belgium. [4]Structural Biology Brussels, Vrije Universiteit Brussel, Pleinlaan 2, Brussels, Belgium. [5]VIB-KU Leuven Center for Cancer Biology, VIB, Herestraat 49, Leuven, Belgium. [6]Department of Oncology, KU Leuven, Herestraat 49, Leuven, Belgium. [7]Research Group of Organic Chemistry, Vrije Universiteit Brussel, Pleinlaan 2, Brussels, Belgium. [8]Jean Jeener NMR Centre, VUB, Brussels, Belgium. ✉e-mail: jan.steyaert@vub.be

one of the protomers but not to the complex will inhibit the interaction.

In this study, we exploited the unique properties of Nbs either to disrupt or to stabilize the SOS1$^{cat}$•KRAS$^{G12V}$ signaling complex (further referred to as SOS1•RAS) to modulate the nucleotide exchange rate of KRAS. RAS GTPases (H-, K-, and N-RAS) control many cellular processes by switching between an inactive GDP-bound state and an active GTP-bound state and represent the most frequently mutated oncoprotein family in human cancers[5,6]. While for many years, KRAS had built a reputation as being undruggable, allele-specific KRAS$^{G12C}$ inhibitors are currently changing the treatment paradigm for patients with KRAS$^{G12C}$-mutated cancer. This success has fueled drug discovery efforts for direct KRAS inhibitors[7]. An alternative approach for modulating RAS activity is to reduce the level of activated RAS by either stabilizing[8] or inhibiting[9] the interaction of RAS with GTPase activating proteins or GDP/GTP exchange factors (GEFs) such as SOS1 (Son of sevenless homolog 1).

For the simultaneous discovery of Nbs that either stabilize or disrupt the SOS1•RAS signaling complex to modulate nucleotide exchange, we devised generic immunization (ChILL) and nanobody selection (DisCO) strategies that are widely applicable to other PPIs. To highlight the power of this platform technology, we describe the discovery of four representative Nbs, including competitive disruptors but also connective and allosteric stabilizers of the SOS1•RAS complex. We show that these Nbs modulate the PPIs and affect the nucleotide exchange rates in vitro and *in cellulo* by different molecular mechanisms. To benchmark our work to the genuine regulatory mechanisms that occur in vivo, we also compared the properties of our allosteric Nbs to the behavior of the allosteric RAS•GTP that binds a second highly conserved RAS binding site on SOS1 and activates SOS1 in a positive feedback loop[10].

## Results and discussion

### Cross-link PPIs and Immunize llamas (ChILL) for eliciting allosteric nanobodies that bind the SOS1•RAS complex

For the in-vivo maturation of allosteric or connective Nbs that bind and stabilize transient protein complexes, one could, in principle, immunize llamas with mixtures of the associating protomers. However, considering the short half-life of transient PPIs, we anticipated that this approach is likely to be highly inefficient. Even if a PPI is stabilized with specific ligands (such as metabolites, ATP, Ca++, etc.), small amounts of the free protomers will induce the maturation of protomer-specific Nbs, many of which act as orthosteric or allosteric inhibitors of the PPI. During progressive rounds of immunization, more and more protomer-specific Nbs will mature and circulate, causing less and less protomers to associate in the complex for triggering the formation of PPI-specific Nbs. To overcome this progressive pitfall, we cross-linked the SOS1•RAS complex with glutaraldehyde to freeze the interacting proteins in a covalent association very similar to the native PPI. Next, we immunized llamas with this antigen to trigger and mature allosteric Nbs that bind conformational epitopes exposed on this stabilized complex (Supplementary Fig. 2). Similarly, we previously applied ChILL for the discovery of allosteric Nbs that stabilize the human β2AR•Gs signalling complex[11] or yeast Vps34 complex II[12], respectively.

### Display and co-selection (DISCO) of nanobodies that modulate the SOS1•RAS complex

ChILL generates different classes of binders including connective and allosteric Nbs that bind the complex, but also competitive and allosteric Nbs that inhibit PPIs. For the enrichment and efficient partitioning of target-specific Nbs that selectively bind to the (non-cross-linked) complex from those that bind to the dissociated protomers, we displayed our ChILL immune libraries on yeast[13,14]. For the co-selection of cells displaying Nbs of different classes, we labeled purified SOS1 and RAS (RAS•GDP) separately with different fluorescent dyes and stained our yeast display libraries with mixtures of these fluorescent protomers (Fig. 1a). Multicolor fluorescent-activated cell sorting (FACS) was then used to separate cells displaying Nbs that only bind to one of the protomers from cells that are stained simultaneously by SOS1 and RAS, indicating that they display a Nb that binds the binary complex (Fig. 1b, Supplementary Fig. 3a). It took only three consecutive rounds of co-selection by multicolor FACS to sort and isolate yeast cells displaying a Nb that either binds to one of the fluorescent proteins separately, or to the complex (Supplementary Fig. 3b). Single yeast clones of the different subsets were then cultured and stained again with fluorescently labeled SOS1 and/or RAS and further analyzed by flow cytometry to classify the displayed Nbs (Supplementary Fig. 4). The most dissimilar binders were sequenced and recloned for expression in *E. coli*. Two competitive binders that inhibit the SOS1•RAS association (Nb77 & Nb84) and two Nbs (Nb14 & Nb22) that bind the binary complex were fully characterized to demonstrate the power of our workflow aimed at discovering diverse Nbs that modulate PPIs by different mechanisms.

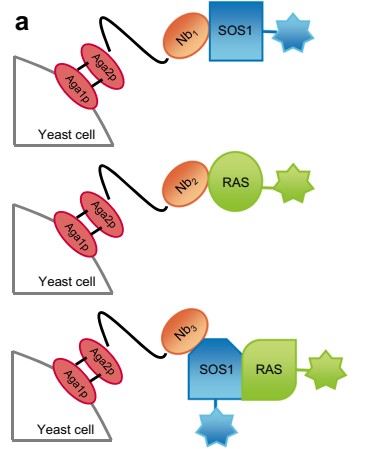
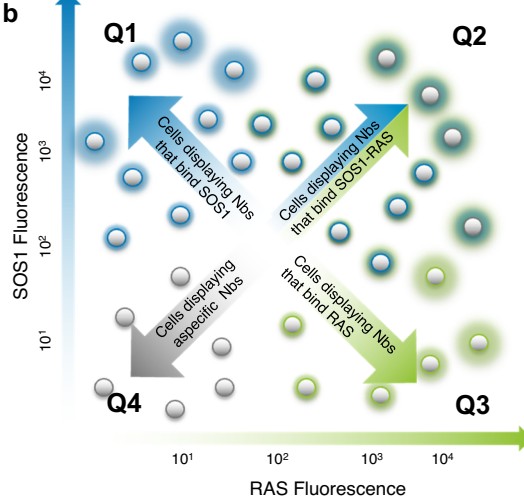

**Fig. 1 | Rationale of DisCO for partitioning target-specific Nbs that selectively bind to the SOS1•RAS complex only from those that only bind to the separate SOS1 or RAS protomers. a** yeast cells displaying the ChILL immune library were incubated with SOS1 and RAS labeled with different fluorescent dyes. **b** FACS can then be used to sort yeast cells that display Nbs that bind to one of the protomers (Q1 or Q3) and others that are stained simultaneously by SOS1 and RAS (Q2). Supplementary Fig. 3 shows how this principle was put into practice to select competitive, connective and allosteric binders of the SOS1•RAS complex.

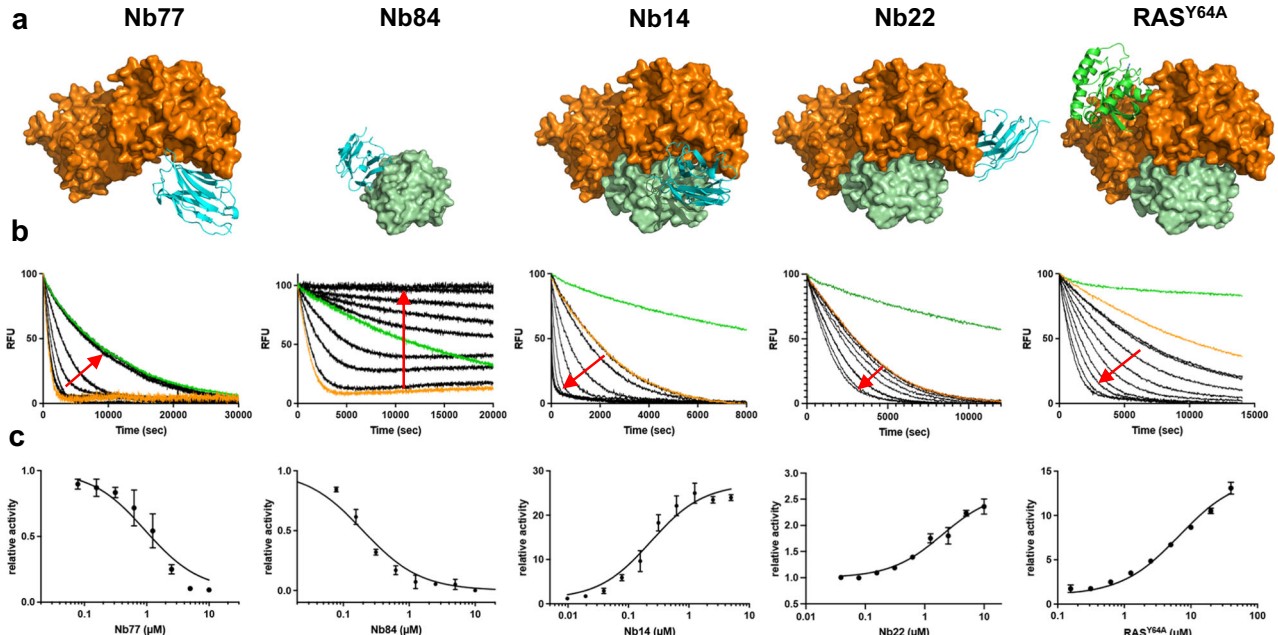

**Fig. 2 | Competitive binders to SOS1 (Nb77) or RAS (Nb84) inhibit nucleotide exchange, while allosteric binders to the SOS1•RAS complex (Nb14 and Nb22) increase nucleotide exchange. a** Crystal structures of the Nbs (cyan, cartoon representation) in complex with SOS1 (orange) and/or RAS (green). The structure of the ternary complex SOS1•RAS•RAS$^{Y64A}$•GppNHp (RAS$^{Y64A}$, cartoon representation) is also shown for comparison (pdb ID 1NVV). **b** SOS1-catalyzed nucleotide exchange assays on RAS performed in presence of the Nbs or RAS$^{Y64A}$ (black curves). Single turnover nucleotide exchange experiments were started by mixing RAS•mGDP (1 µM) with an excess of GTP (200 µM) at a fixed concentration of SOS1 (0.05, 0.3 or 1 µM depending on the allosteric binder). The red arrows follow increasing concentrations of the Nbs or RAS$^{Y64A}$. As references, we included the intrinsic nucleotide exchange of RAS alone (green curves) and the nucleotide exchange of RAS in complex with SOS1 (orange curves). **c** Single turnover time traces were fitted as single exponential decays and the apparent rates were normalized to the SOS1-catalyzed nucleotide exchange rate (orange curves) and plotted as a function of the nanobody or RAS$^{Y64A}$ concentration. Data are presented as mean values +/− SEM. Nb77 fully inhibits SOS1 activity (SOS1 at 1 µM, EC50 of 0.95 µM ± 0.1 µM). Nb84 fully inhibits SOS1 activity and abolishes the intrinsic nucleotide exchange reaction of RAS (SOS1 at 1 µM, EC50 of 0.20 ± 0.02 µM). Nb14 increases SOS1 activity 27-fold (SOS1 at 0.05 µM, EC50 of 0.23 µM ± 0.03 µM). Nb22 increases SOS1 activity 2.6-fold (SOS1 at 1 µM, EC50 of 2.0 µM ± 0.3 µM). The allosteric RAS increases SOS1 activity 14-fold (SOS1 at 0.3 µM, EC50 of 6.7 ± 0.5 µM). The experiments were repeated independently three times. Source data are provided as a Source Data file.

## Competitive binders to SOS1 or RAS inhibit nucleotide exchange

Nb77 was derived from the Q1 subset of yeast cells (Supplementary Fig. 3b) and binds to SOS1 only but not to RAS, as supported by flow cytometry (Supplementary Fig. 4) and Bio-Layer Interferometry (BLI) (Supplementary Fig. 5). A high-resolution crystal structure of SOS1 in complex with Nb77 (Fig. 2, Supplementary Table 1) confirms that Nb77 binds to the edge of the Cdc25 domain of SOS1 and occupies part of the catalytic RAS binding site, resulting in the full dose-dependent inhibition of the SOS1-catalyzed nucleotide exchange reaction (Fig. 2, Supplementary Fig. 6a).

Reciprocal to Nb77, Nb84 was collected from the Q3 subset of yeast cells (Supplementary Fig. 3b). Nb84 binds to the switch-1 loop of RAS (KRAS$^{G12V}$•GDP) (Fig. 2, Supplementary Figs. 4, 5), preventing interactions with SOS1 and causing competitive inhibition of the SOS1-catalyzed nucleotide exchange reaction (Fig. 2, Supplementary Fig. 6b). Remarkably, this RAS-specific Nb further inhibits the intrinsic nucleotide exchange of RAS, locking it in a fully inactive conformation. We hypothesize that the interactions of Nb84 with the switch-1 loop of RAS prevent the conformational changes required to exchange the nucleotide, very similar to the conformation-locking antibodies discovered by the Evangelista lab[15].

## Allosteric binders stabilize the SOS1•RAS complex and accelerate nucleotide exchange

Nb14 and Nb22 were collected from the Q2 subset of yeast cells that were co-selected for the simultaneous binding of RAS and SOS1 (Fig. 1, and Supplementary Fig. 3b), and BLI experiments confirmed that these Nbs can bind to the SOS1•RAS complex but also to SOS1 alone (Supplementary Fig. 5). Next, we solved the structures of the corresponding SOS1•RAS•Nb ternary complexes by X-ray crystallography (Fig. 2, and Supplementary Table 1) and found that Nb14 is a connective binder to the SOS1•RAS heterodimer, in which CDR2 and CDR3 interact with SOS1, whereas CDR1 interacts primarily with RAS (Supplementary Fig. 7). Similar to the inhibitory Nb77, Nb14 interacts with the Cdc25 domain of SOS1, but Nb14 adopts an orientation that does not cause any steric hindrance for the binding of RAS to the catalytic site. Rather than blocking the protein-protein interaction, Nb14 stabilizes the interaction and accelerates SOS1-catalyzed nucleotide exchange at least 27-fold (Fig. 2).

Nb22 is a fully allosteric antibody that binds to a different epitope on the Cdc25 domain of SOS1, opposite to the catalytic site where RAS and Nb14 bind (Fig. 2). Almost all the interactions of Nb22 with SOS1 are established through framework residues of the antibody, except the R101 side chain of CDR3 that is involved in a salt bridge with RAS-E107 (Supplementary Fig. 8). Remarkably, this allosteric Nb also increased the SOS1-catalyzed nucleotide exchange rate at least 2.6-fold (Fig. 2).

## Nb14 and RAS•GTP modulate SOS1 activity by a similar allosteric activation mechanism

Much data support the view that SOS1 has evolved an allosteric activation mechanism that extends beyond the process of membrane recruitment, and it has been shown that SOS1 activity is regulated by a positive feedback mechanism triggered by a second GTP-bound RAS monomer that binds an allosteric site on SOS1, opposite to its catalytic site[10]. KRAS$^{Y64A}$ bound to GppNHp (further referred to as RAS$^{Y64A}$) can be used to monitor the effects of this allosteric RAS because the Y64A mutation prevents RAS from preferentially binding the catalytic site on

**a**

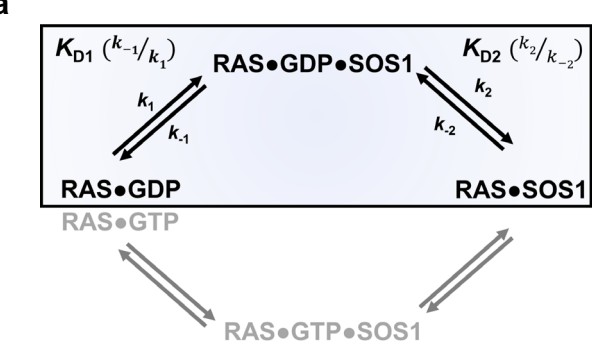

**b**

| | SOS1 | SOS1 + Nb22 | SOS1 + Nb14 | SOS1 + RAS$^{Y64A}$ |
|---|---|---|---|---|
| $k_1$ ($k_{on}$) ($10^6$ M$^{-1}$s$^{-1}$) | 4.2 | 15.8 | 0.34 | 0.26 |
| $k_{-1}$ ($k_{off}$) (s$^{-1}$) | 73 | 88 | 20 | 56 |
| $K_{D1}$ ($10^{-6}$ M) | 17.5 | 5.6 | 57.4 | 219 |
| $k_2$ ($k_{off}$) (s$^{-1}$) | 0.003 | 0.003 | 1.422 | 0.671 |
| $k_{-2}$ ($k_{on}$) ($10^6$ M$^{-1}$s$^{-1}$) | 0.00045 | 0.00028 | 0.052 | 0.113 |
| $K_{D2}$ ($10^{-6}$ M) | 6.7 | 10.7 | 26.8 | 6 |

**c**

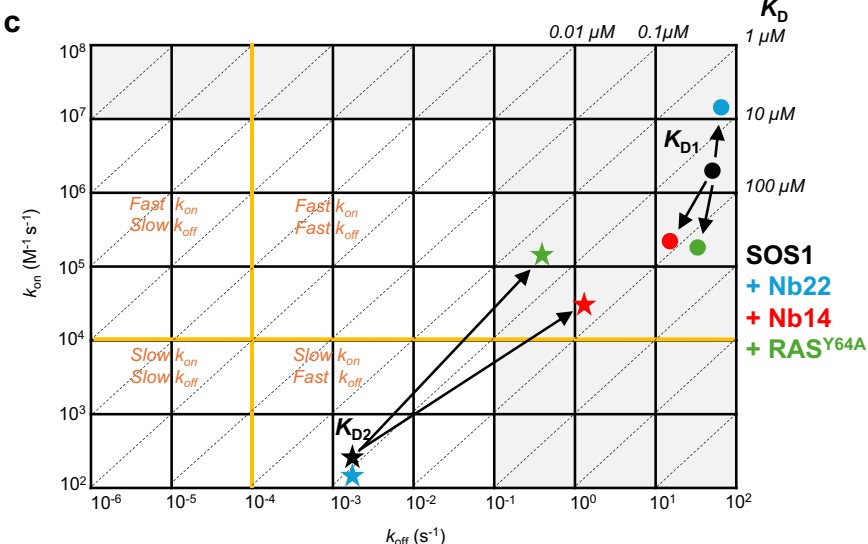

**Fig. 3 | The allosteric binders Nb14, Nb22 and RAS$^{Y64A}$ impact the SOS1-RAS nucleotide exchange cycle by different mechanisms. a** Minimal four-step mechanism describing the SOS1-catalyzed nucleotide exchange cycle of RAS including the formation of the transition state RAS•GDP•SOS1 (top) followed by nucleotide dissociation and accumulation of the more stable RAS•SOS1 complex. The cycle is completed by the binding of GTP to form RAS•GTP•SOS1, and the subsequent release of SOS1, causing the accumulation of RAS•GTP. Because of the symmetry of this cycle, we only determined the kinetic parameters of the boxed part. **b** Effects of the allosteric nanobodies and allosteric RAS on the kinetics of the SOS catalyzed nucleotide exchange reaction. **c** Kinetic map to visually compare the effects of the different allosteric modulators on the nucleotide exchange kinetics and equilibria.

SOS1, whereas the non-hydrolysable GTP analog (GppNHp) locks this allosteric modulator in its activated state[16]. Assayed under the same conditions as Nb14 & Nb22, RAS$^{Y64A}$ increased SOS1 activity 14.4-fold (Fig. 2). It thus appears that our allosteric Nbs increase SOS1-catalyzed nucleotide exchange to a similar extent than the natural allosteric modulator.

To further interrogate if Nb14, Nb22 and RAS•GTP modulate SOS1 activity via a similar allosteric activation mechanism, we compared the effects of Nb14, Nb22 and RAS$^{Y64A}$ on the reaction coordinate of the complete nucleotide exchange cycle. The SOS1-catalyzed nucleotide exchange reaction is a complex multi-step process involving the binding of RAS•GDP to the catalytic site of SOS1 ($K_{D1} = k_{-1}/k_1$), followed by the release of GDP to yield a nucleotide-free SOS1•RAS complex ($K_{D2} = k_2/k_{-2}$) (Fig. 3a). This binary complex is susceptible to the binding of GTP, causing the dissociation of SOS1 and the accumulation of RAS•GTP to finish the exchange reaction. As studies have shown that GEFs do not impose a preference on whether the GTPase is reloaded with GTP or GDP, the complete nucleotide exchange cycle can conveniently be simplified as a two-step mechanism with GDP only[17]. Accordingly, we compared the effects of excess amounts of Nb14 or Nb22 to the impact of allosteric RAS on the reaction coordinate of the SOS1-catalyzed nucleotide exchange reaction of RAS. Following the same

assumptions and simplifications proposed by Roger S. Goody and co-workers[18] to study the Ran/RCC1 system, we similarly performed single turnover experiments in a stopped-flow fluorimeter under two pseudo-first-order conditions to measure all the microscopic rate constants[18]. In a first experiment, we mixed 0.5 μM of the RAS•mGDP substrate with varying concentrations of SOS1 and an excess of GDP to determine $K_{D1}$ and $k_2$. In the reverse direction, we also mixed 0.5 μM of SOS1•RAS with increasing amounts of mGDP to measure $K_{D2}$ and $k_{-1}$ (Supplementary Fig. 9). These experiments confirm that the formation of the SOS1•RAS•GDP ternary complex is fast at the concentrations used in our set-up ($k_1$, $4.2 \times 10^6$ M$^{-1}$s$^{-1}$), while the transition from the ternary complex to the nucleotide free complex represents the rate-limiting step of the exchange reaction ($k_2$, 0.003 s$^{-1}$) corresponding to a half-life of the ternary complex of almost 4 min, incompatible with a tight regulation of RAS (Fig. 3b). Our extensive kinetic analysis shows that Nb14 modulates the SOS1-catalyzed exchange reaction in a way that is very similar to the allosteric RAS, but different to Nb22. Indeed, Nb14 and RAS$^{Y64A}$ decelerate the formation of the ternary complex ($k_1$) to the same extent (-12−16x), while Nb22 accelerates its formation (~4x). Moreover, Nb14 and RAS$^{Y64A}$ have a major impact on the rate limiting nucleotide release step and increase $k_2$ 474 and 224-fold respectively, while Nb22 is neutral to this process (Fig. 3c).

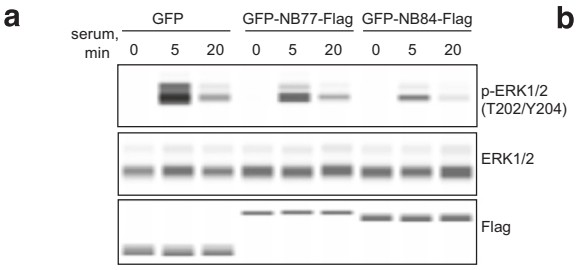

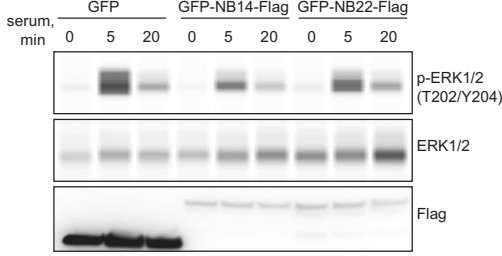

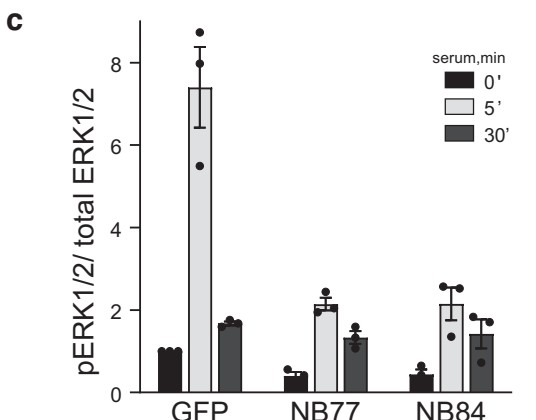

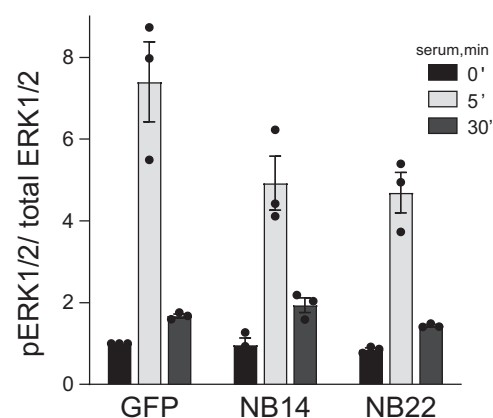

**Fig. 4 | Nb14, 22, 77 and 84 inhibit RAS signaling through the Raf/MEK/ERK pathway as assessed by ERK1/2$^{T202/Y204}$ phosphorylation levels. a, b** HEK293T cells expressing GFP or the indicated Nbs were serum-starved, stimulated with 10% serum, and analyzed by WES capillary immunoassay. **c, d** ERK1/2 phosphorylation. Values are means of phosphorylated (p) relative to non-phosphorylated protein levels ± SEM for (**a**, **b**), respectively. The experiments were repeated independently three times. Source data are provided as a Source Data file.

Kuryan and coworkers compared the structure of SOS1 alone[16] to the structures of the binary SOS1•RAS complex[19] and the ternary SOS1•RAS•RAS$^{GTP}$ complex[10] and observed that the binding of RAS to the catalytic site of SOS1 requires the helical hairpin of the cdc25 domain to be pulled back. They proposed that allosteric binding of RAS•GTP to SOS1 facilitates the rotation and opening of the helical hairpin, opening up the catalytic site to bind RAS[20]. This model implies that the conformational changes in SOS1 happen before RAS•GDP binding, to produce a binding-competent state (conformational selection) and not in response to ligand binding (induced fit). Our kinetic data, however, indicate that the binding of the allosteric RAS•GTP or Nb14 slows down the bimolecular association of SOS1 and RAS•GDP (at least 10-fold) but accelerates nucleotide release by two orders of magnitude. These data hint to an induced fit mechanism where conformational changes in the SOS1•RAS•GDP ternary complex, leading to the rotation and opening of the helical hairpin, occur in response to the binding of RAS•GDP. The catalytic power of SOS1 can then be explained by a better structural complementarity between SOS1 and nucleotide free RAS, relative to the ternary complex. The binding of Nb14 or the allosteric RAS$^{Y64A}$ further accelerates nucleotide exchange by stabilizing the same 'open' conformation of the SOS1•RAS complex, in which nucleotides can diffuse in and out with less steric barriers. Consistent with this notion, Nb14 has been affinity-matured by multiple immunizations with the nucleotide free binary complex.

### Allosteric nanobodies for target validation
PPIs are an attractive emerging class of molecular targets and are critically important in the progression of many diseases[21] but validating them as druggable targets remains a major challenge. One key advantage of Nbs over other antibodies is that they can be expressed as intrabodies, allowing us to monitor their effect on signaling pathways and cell metabolism inside living cells[22]. This raised the question if our PPI-modulating Nbs could be used for target validation by interrogating relevant preclinical models for effects on the specific disease. Therefore, we assessed the potential of our Nbs to alter the RAS/MAPK signaling pathway by measuring ERK1/2$^{T202/Y204}$ phosphorylation levels (Fig. 4). As expected, ectopic expression of Nb77 and Nb84, which inhibit SOS1•RAS nucleotide exchange, led to lower levels of ERK phosphorylation after serum stimulation. Interestingly, Nb14 and Nb22 which increase the nucleotide exchange rate also substantially decreased the level of ERK1/2 phosphorylation. However, this seemingly contradictory result is completely in line with previously developed small molecule compounds that bind and activate SOS1 in a similar way (see below). The decreased levels of Erk phosphorylation could be due to a biphasic modulation of RAS•GTP and phospho-ERK levels through negative feedback loops[23].

### Nanobodies as leads for allosteric peptidomimetics
Antibodies emerged as clinically useful drugs but also have their limitations related to size and cell penetration. These obstacles can be circumvented by the design of peptides mimicking the CDRs[24], with Nbs having the advantage of presenting a long CDR3 often penetrating antigen cavities[25]. Accordingly, we synthesized peptidomimetics of the CDR3 of the most potent Nb14. Several rounds of optimization by rational design allowed us to generate potent peptides sharing similar in vitro and *in cellulo* effects compared to the original Nb, thereby providing the first proof-of-concept of Nb-based peptidomimetics able to functionally modulate a protein-protein interaction[26].

### PPI-stabilizing nanobodies to facilitate precision drug discovery
Because many Nbs act as structural chaperones that stabilize specific conformations or interactions, they facilitate small molecule/

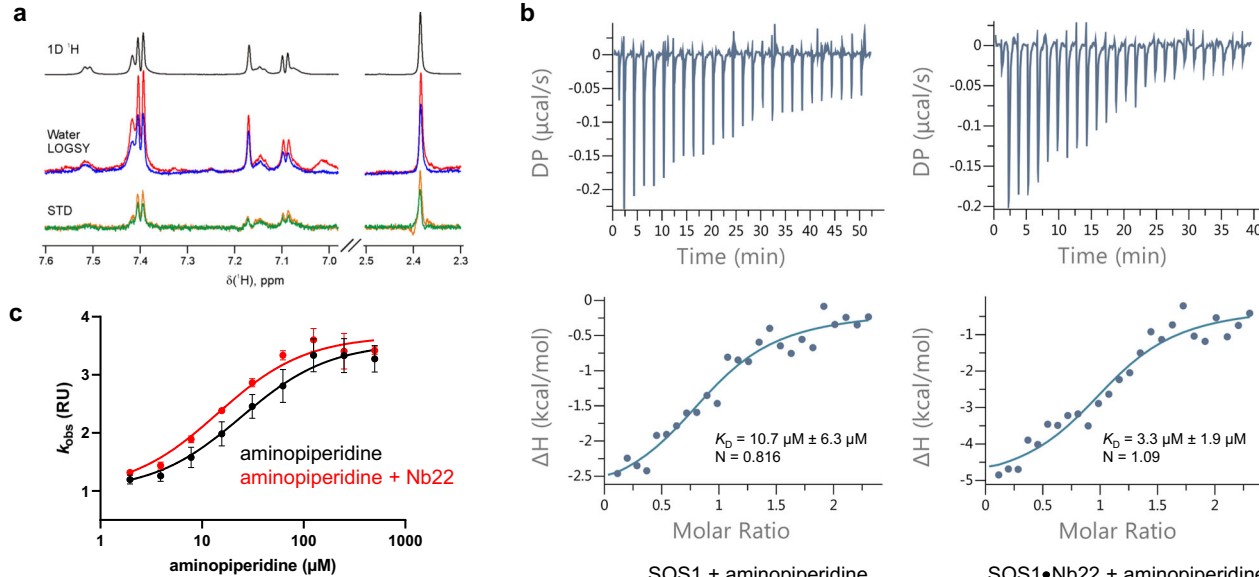

**Fig. 5 | Nb22 increases the affinity of the aminopiperidine indole ligand to SOS1•RAS as shown by NMR spectroscopy, ITC and nucleotide exchange assays. a** NMR spectra measuring the binding of the aminopiperidine indole, an allosteric SOS1•RAS compound discovered by Burns et al.[8], to the SOS1•RAS complex in the presence or absence of Nb22. The different plots show several regions of the 1D 1H reference spectrum of aminopiperidine indole (black), water ligand observed via gradient spectroscopy (water-LOGSY) spectra of aminopiperidine indole in the presence of SOS1·RAS alone (blue) and with 4 molar equivalents of Nb22 (red), and saturation transfer difference (STD) spectra of the compound in the presence of SOS1·RAS alone (green) and with 4 molar equivalents of Nb22 (orange). Estimated from the WaterLOGSY spectra, Nb22 enhances the binding signal of aminopiperidine indole to SOS1·RAS by a factor of 2. All experiments were performed in 20 mM potassium phosphate 75 mM NaCl pH 7.0 at 298 K. **b** ITC

experiments to measure the affinity of SOS1 for the aminopiperidine indole in presence or absence of Nb22. In each case, the sample cell was filled with 30 μM of SOS1 in the presence (50 μM) or absence of Nb22 while the syringe was loaded with 300 μM of ligand. **c** SOS1·RAS nucleotide exchange assay performed with increasing concentrations of aminopiperidine indole in the presence (red) or absence (black) of an excess of Nb22. Data are presented as mean values +/− SEM. In the presence of Nb22, the maximum activation of SOS1 by aminopiperidine is unchanged (3.5 to 3.7-fold) but the apparent EC50 is lowered approximately two fold. The best fit value of the EC50 is 24.2 μM with the 95% confidence interval ranging from 18.5 μM to 31.6 μM in presence of aminopiperidine and 14.2 μM with the 95% confidence interval ranging from 11.7 μM to 17.2 μM in presence of aminopiperidine and Nb22. The experiments were repeated independently three times. Source data are provided as a Source Data file.

fragment library screenings by enhancing the binding of (weak) ligands that bind the same conformer[15,27,28]. Previously, a series of indolo-4-aminopiperidines were identified as compounds that bind the SOS1•RAS interface and elicit the acceleration of SOS1-catalyzed nucleotide exchange[8]. Here we show that addition of Nb22 to SOS1•RAS•aminopiperidine indole leads to a marked increase in the signal intensity in solution NMR experiments ubiquitously used in ligand-based fragment screening (Fig. 5a). The observed nanobody-mediated signal enhancement is especially valuable for the screening of fragment libraries by NMR, a common first step in fragment-based drug discovery. Consistent with the NMR data, we performed isothermal titration experiments and measured a 3-fold increase of the affinity of this compound for the SOS1•Nb22 complex, relative to SOS1 alone (Fig. 5b). As expected from positive allosteric modulators that bind different allosteric sites, we also observe that Nb22 and aminopiperidine synergize the acceleration of the SOS1-catalyzed nucleotide exchange (Fig. 5c). It thus appears that our methods enable the discovery of Nbs that bind druggable targets in conformations amenable to small molecule drug screening and design.

In conclusion, we showed that ChILL and DisCO can be combined for the discovery of inhibiting, connecting or allosteric nanobodies that stabilize or disrupt PPIs. These in vivo matured nanobodies can then be used for extensive biochemical, functional and structural characterization of transient PPIs and for their validation as a drug target. Because many of these nanobodies bind druggable cavities on- or stabilize druggable conformations of these protein complexes, these nanobodies may serve as starting points for structure-based drug design or may facilitate precision drug discovery by fragment or

compound screening on a target class that has long been considered as undruggable.

## Methods

### Recombinant protein expression and purification

Codon-optimized synthetic genes (GenScript) encoding the human KRAS[G12V] and KRAS[Y64A] mutants (residues 1–169) and human SOS1 (residues 564–1049) were cloned as NdeI and XhoI fragments into pET28b (Novagen). His-tagged KRAS•GDP (abbreviated RAS) and SOS1 were expressed in *E. coli* BL21 DE3 and purified by immobilized metal ion affinity chromatography (IMAC) and size-exclusion chromatography as described previously[1]. The His-tag was either kept or cleaved off by TEV protease (12 h at 4 °C), after which the uncleaved protein was removed by a second IMAC and conserved in a buffer containing 25 mM Hepes (pH 7.5), 150 mM NaCl and 10 mM MgCl₂. RAS loaded with mant•GDP (RAS•mGDP) was obtained by mixing RAS with a 10x stoichiometric excess of mGDP (Jena Bioscience) in a buffer containing an excess (20 mM) of EDTA. After overnight incubation at 4 °C, an excess of MgCl2 was added (50 mM) for 30 min and the free nucleotides were removed by gel filtration. RAS loaded with GppNHp was obtained by mixing RAS with a 10x stoichiometric excess of GppNHp (Jena Bioscience) in a buffer containing an excess (20 mM) of EDTA and alkaline phosphatase (Roche). After overnight incubation at 4 °C, an excess of MgCl2 was added (50 mM) for 30 min and RAS[Y64A]•GppNHp was purified by gel filtration. SOS1•RAS complex was formed by incubating SOS1 and RAS at a stoichiometric ratio of 1:3 overnight at 4° with 20 mM EDTA and alkaline phosphatase (Roche). The complex was then purified by gel filtration. The ternary complex RAS•SOS1•RAS[Y64A] (RAS[G12V]•SOS1•RAS[Y64A]•GppNHp) was obtained as described in ref. 10.

The open reading frames of nanobodies selected by yeast display were cloned into a Golden Gate variant of pMESy4, expressed in *E. coli* strain WK619 and purified as described in ref. 29.

### Nanobody generation and selection

For crosslinking, proteins were incubated with glutaraldehyde (0.001% final concentration) for 1 h at room temperature. Once the complex band was clearly observed, the reaction was scaled up (to 1 mg) for llama immunization. The crosslinked material was aliquoted and frozen for llama immunization according to published methods[11]. The nanobody library for yeast display was constructed according to ref. 14. Yeast display libraries were inoculated, induced and orthogonally stained with CoA-Alexa547 as described in ref. 14. In each round of selection, $4 \times 10^7$ stained yeast cells were incubated for 60 min at 4 °C (rotating at 50 rpm) with the corresponding fluorescent antigen at 100 nM. After incubation with the fluorescent antigens, yeast cells were washed three times with a buffer (Hepes 25 mM pH 7.5, NaCl 150 mM, MgCl2 10 mM) and resuspended in a 2 ml final volume for sorting on a FACS Aria (BD Biosciences). To optimize the selection, yeasts were gated based on their SSC-A and FSC-A profile as single cells and then as expressing nanobodies. Finally, depending on the desired output, selections were performed using quadrants Q1 (SOS1-488 binders), Q2 (both SOS1-488 and RAS-647 binders) or Q3 (RAS-647 binders). Selected yeast cells were sorted into SDCAA medium, grown and induced, then stained again for consecutive rounds of selection.

### Nanobody screening by flow cytometry

Sorted yeast clones were grown separately in 96-well plates and induced in 500 µl SGCAA. Cells displaying a well-characterized nanobody that binds an irrelevant antigen were included systematically as a negative control. For flow cytometry analysis of individual clones, $10^6$ cells were routinely transferred to a fresh 96-well plate, washed and stained orthogonally with CoA-647 in a final volume of 15 µl. $10^5$ of these stained cells were mixed with serial dilutions (6, 12, 25, 50 nM) of the fluorescent antigens in 50 µl of buffer (Hepes 25 mM pH 7.5, NaCl 150 mM, MgCl2 10 mM) and incubated for 60 min at 4 °C with shaking 50 rpm. To remove unbound fluorescent antigens, cells were washed two times with the same buffer and applied on a FACS Fortessa (BD Biosciences). Routinely, 10,000 yeast cells of each well were analyzed using FlowJo software (FlowJo, LLC) and compared to the negative control. Sequences were obtained by Sanger sequencing.

Nb77:(QVQLVESGGGLVQPGGSLRLSCAASRSISSINIMGWYRQAP GKERESVASHTRDGSTDYADSVKGRFTISRDNAKNTVYLQMNSLKPED TAVYYCTTLTGFPRIRS)

Nb84:(QVQLVESGGGLVQAGESLRLSCAASVSIFSINTMGWYRQAP GKPRELVARIFTGGSTYYVDSVKGRFTISRDNAKNTVYLQMNQLKPEDT GVYYCRLGADY)

Nb14:(QVQLVESGGGLVQAGGSLRLSCAASRSSFTINRMGWYRQAP GKQRELVADITSGGNRNYADSVKGRFTIARDNAKNTAYLQMNSLKPEDT AVYYCNAKIHPWSVADL)

Nb22:(QVQLVESGGGLVQAEGSLRLSCLVSGDIVRSNLMGWYRQAP GKQREFVARINPTGSANYADSVRGRFTISKDNSKKTLYLQMGSLQPEDT AVYYCRLIQNRDY)

Nb75:(QVQLVESGGGLVQAGGSLRLSCAASGNILPINVMGWYRQTP GSQRELVATIVTSGGSTAGNTNYVDSVKGRFTISGDNAKNTVYLQMSS LKPEDTAVYYCNLKTRRAPWATPNNY)

### Protein complexes crystallization and data collection

All protein complex crystals were obtained by the sitting-drop vapor-diffusion method at 293 K. Before being flash frozen in liquid nitrogen, all crystals were briefly soaked in mother liquor supplemented with 20% glycerol. SOS1•Nb77 complex was obtained by mixing SOS1 and Nb77 at a stoichiometric ratio of 1:2 for 1 h at 4 °C. The complex was then purified by gel filtration to remove the excess of free nanobody and concentrated to 15 mg/ml. Crystals were obtained in the presence of 2 M ammonium sulfate and 0.1 M Tris, pH 8.5 (Index screen condition A6, Hampton Research). Data were collected at 100 K on the I03 beamline at the Diamond Light Source synchrotron (Oxfordshire, UK) and the structure was refined to 1.9-Å resolution. The complex crystallized in the space group P21 21 21 with a single protein complex in the asymmetric unit (PDB ID 8BE2). RAS•Nb84 complex was obtained by mixing RAS (KRAS$^{G12V}$•GDP) and Nb84 at a stoichiometric ratio of 1:2 for 1 h at 277 K. The complex was then purified by gel filtration to remove the excess of free nanobody and concentrated to 30 mg/ml. Crystals were obtained in the presence of 20% PEG 4000 and 0.1 M sodium cacodylate, pH 6.0 (Proplex screen condition B9, Molecular Dimensions). Data were collected at 100 K on the Proxima 1 beamline of the Soleil synchrotron (Saint-Aubin, France) and the structure was refined to 1.84-Å resolution. The complex crystallized in the space group P1 21 1 with two protein complexes in the asymmetric unit (PDB ID 8BE3). SOS1•RAS•Nb14 complex was obtained by mixing SOS1, RAS and Nb14 at a stoichiometric ratio of 1:2:2 with an excess of EDTA (20 mM final) for 1 h at 4 °C. The complex was then purified by gel filtration to remove the excess of free molecules and concentrated to 30 mg/ml. Crystals were obtained in the presence of 20% PEG 1000 and 0.1 M Tris, pH 7.0 (Wizard screen condition B7, Hampton Research). Data were collected at 100 K on the on the I03 beamline at the Diamond Light Source synchrotron (Oxfordshire, UK) and the structure was refined to 1.9-Å resolution. The complex crystallized in the space group I4 with a single protein complex in the asymmetric unit (PDB ID 8BE4). SOS1•RAS•Nb22•Nb75 complex was obtained by mixing SOS1, RAS, Nb75 and Nb22 at a stoichiometric ratio of 1:2:2:2 with an excess of EDTA (20 mM final) for 1 h at 4 °C. The complex was then purified by gel filtration to remove the excess of free molecules and concentrated to 20 mg/ml. Nb75 was used as a crystallization chaperone because no crystals of SOS1•RAS•Nb22 diffracted at high resolution. Crystals were obtained in the presence of 10% PEG 4000, 0.1 M sodium citrate, pH 5.5 and 0.2 M sodium acetate (Proplex screen condition B3, Molecular Dimensions). Data were collected at 100 K on the on the I24 beamline at the Diamond Light Source synchrotron (Oxfordshire, UK) and the structure was refined to 3.13-Å resolution. The complex crystallized in the space group I4 2 2 with a single protein complex in the asymmetric unit (PDB ID 8BE5).

### Structure refinement and analysis

All diffraction data were integrated and scaled with XDS[30]. Models were built by iterative cycles of refinement with Phenix[31] and manual building in Coot (Emsley, 2010). MolProbity was used for structure validation[32]. Ramachandran statistics (% Favored:% Outlier) were 98.4:0.0 (PDB ID 8BE2), 99.1:0.0 (PDB ID 8BE3), 98.3:0.0 (PDB ID 8BE4), and 89.1:1.0 (PDB ID 8BE5). Data collection and refinement statistics are summarized in Supplementary Table 1. Figures were produced with PyMOL (http://www.pymol.org/).

### Biolayer Interferometry (BLI) experiments

Biolayer interferometry measurements were performed using Streptavidin biosensors on an Octet Red96 (Forte Bio, Inc.) system at 25 °C, shaking at 1000 rpm and in a buffer containing 25 mM Hepes pH 7.5, 150 mM NaCl, 10 mM MgCl2, 0.1% BSA and 0.005% Tween 20. The protein immobilized on biosensors was previously biotinylated using the EZ-Link NHS-Biotin reactant (ThermoScientific). Following a first step of equilibration with the buffer, the biosensors were loaded with biotinylated protein at a concentration of 1 µg/mL to reach an amplitude signal of 1. After baseline acquisition, the biosensors were transferred to wells supplemented with increasing concentrations of the protein of interest to measure the association step. Finally, the biosensors were transferred to wells containing the buffer to measure the dissociation step. All samples were prepared and measured in triplicate. To obtain an apparent dissociation constant, the amplitude signal at equilibrium was plotted *vs.* the protein concentration. The resulting

titration curve was fitted on a Langmuir equation. To study the affinity of the nanobodies for SOS1 and RAS, biotinylated nanobodies were loaded on sensors and plunged into wells containing varying concentrations (0 to 25 µM) of SOS1, RAS or SOS1•RAS.

### SOS1-catalyzed nucleotide exchange assays

SOS1-catalyzed nucleotide exchange assays were conducted measuring mGDP fluorescence (440 nm) on a microplate reader (Tecan). The reaction was started by mixing mGDP-loaded RAS (1 µM) with SOS1 (from 0.05 to 3 µM depending on the reaction rate) and an excess of GTP (200 µM). To determine the effect of Nb84, GEF assays were performed in presence of 1 µM of RAS-mGDP, 3 µM of SOS1, 200 µM of GTP and a varying concentration of Nb84 (0 to 10 µM). To determine the effect of Nb77, GEF assays were performed in presence of 1 µM of RAS-mGDP, 3 µM of SOS1, 200 µM of GTP and a varying concentration of Nb77 (0 to 10 µM). To determine the effect of Nb14, GEF assays were performed in presence of 1 µM of RAS-mGDP, 0.05 µM of SOS1, 200 µM of GTP and a varying concentration of Nb14 (0 to 10 µM). To determine the effect of Nb22, GEF assays were performed in presence of 1 µM of RAS-mGDP, 1 µM of SOS1, 200 µM of GTP and a varying concentration of Nb22 (0–10 µM). To determine the effect of RAS$^{Y64A}$, GEF assays were performed in presence of 1 µM of RAS-mGDP, 0.3 µM of SOS1, 200 µM of GTP and a varying concentration of RAS$^{Y64A}$ (0–40 µM). Raw fluorescence data were fitted to a single exponential decay function. EC$_{50}$ values were calculated by plotting derived rates as a function of ligand concentration and fit using a four-parameter dose–response curve using Prism 8 (Graph pad Software Inc.).

### Kinetic measurements of the SOS1-RAS catalytic cycle

$K_1$ and $k_2$ values were determined by measuring mGDP fluorescence (Jena Bioscience) (excitation at 350 nm, emission filtered at 400 nm) on a stopped-flow apparatus (Applied Photophysics). The reaction was started by mixing mGDP-loaded RAS (0.5 µM) with SOS1 (varying concentrations from 0 to 50 µM) and an excess of GDP (100 µM). The ligand (nanobodies or RAS$^{Y64A}$) was added in excess compared to SOS1 concentration (from 5 µM minimum to 1.5x SOS1 concentration for the highest concentrations). Raw fluorescence data were fitted to a single exponential decay function. $K_1$ and $k_2$ values were calculated by plotting derived rates as a function of ligand concentration and fit using the equation indicated in Supplementary Fig. 9 and using Prism 8 (Graph pad Software Inc.). $K_2$ and $k_{-1}$ values were determined by measuring mGDP fluorescence (Jena Bioscience) (excitation at 350 nm, emission filtered at 400 nm) on a stopped-flow apparatus (Applied Photophysics). The reaction was started by mixing SOS1-RAS (0.5 µM) with mGDP (varying concentrations from 0.5 to 25 µM). To measure the effect of the ligand, the same experiment was performed using pre-purified complexes SOS1•RAS•Nb14 / SOS1•RAS•Nb22 / SOS1•RAS•RAS$^{Y64A}$. Raw fluorescence data were fitted to a single exponential association function. $K_2$ and $k_{-1}$ values were calculated by plotting derived rates as a function of mGDP concentration and fit using the equation indicated in Supplementary Fig. 9 and using Prism 8 (Graph pad Software Inc.).

### Recloning of nanobody for intracellular expression in mammalian cells as intrabodies

To clone and express nanobodies in mammalian cells as intracellular intrabodies with C-terminal HA-tag, open reading frames (ORFs) encoding nanobodies (residues 1–128 according to IMGT numbering) were amplified by PCR with primers TU148 (CTAGAGGATCCACCGG TCGCCACCATGCAGGTGCAGCTGGTGGAGTCTGG) and TU149 (GAT TGTCGACTTAAGCGTAATCTGGAACATCGTATGGGTAGCTGGAGAC GGTGACCTGGGT), and cloned as a BamHI/SalI fragment in the pLenti CMV GFP Puro (Addgene plasmid # 17448) to replace an ORF of GFP.

### WES capillary immunoassay

HEK293 cells were seeded on a 6-well plate ($2 \times 10^5$ cells per well) and were cultured in low-glucose Dulbecco's Modified Eagle Medium (Thermo Fisher Scientific) supplemented with 10% FBS and 1% penicillin-streptomycin. GFP-coding or GFP/Flag-tagged nanobody constructs optimized for mammalian cell expression were transfected using the GeneJuice transfection reagent (Merck). 24 h after transfection, the cells were serum-starved overnight. The serum stimulation was performed with low glucose DMEM containing 10% FBS with the indicated time points. The cells were lysed with ice-cold IP lysis buffer (Thermo Fisher Scientific) and centrifuged for 10 min at 4 °C. Protein concentrations were determined using Pierce BCA Protein Assay Kit (Thermo Fisher Scientific). The protein expression levels were measured using the Wes Simple Western system (BioTechne) per the manufacturer's protocol. Briefly, the protein lysate was mixed with IP lysis buffer, 5X fluorescent master mix (Biotechne), and 400 mM DTT (BioTechne) to achieve the final concentration of 0.8 µg/µl protein. The protein lysate was denatured at 95 °C for 5 min. The following primary antibodies were used: rabbit polyclonal phosphor-ERK1/2 (1:50, Cell Signaling #9101), mouse monoclonal ERK1/2 (clone #L34F12, 1:50, Cell Signaling #4696), and mouse monoclonal Flag antibody (clone M2, 1:1000, Merck). The following secondary antibodies were used: anti-rabbit HRP (ready-to-use, Biotechne, DM-001), and anti-mouse HRP (1:50, Cytiva). Protein expression levels were quantified using Compass for SW (version 3.1.8, Biotechne). The digital blot images were obtained using Compass for SW.

### Synthesis of the indolo-4-aminopiperidine small molecule modulator

The reference allosteric small molecule modulator (see below) was synthesized with a purity >98%, as reported previously in ref. 8 (compound no 3). Briefly, the tert-butyl piperidin-4-ylcarbamate (i) and 1H-indole-3-carbaldehyde (ii) (1.1 equiv.) were mixed in dichloromethane for 3 h, followed by reduction overnight using NaHB(OAc)$_3$, giving the intermediate (iii) with a 28% yield after purification by column chromatography. In a second step, the Boc protecting group was removed with trifluoroacetic acid (10% in volume in dichloromethane) overnight giving the compound (iv) in qualitative yield, used without further purification. Finally, the latter was coupled with Boc-Trp-OH (v) (1 equiv.) using PyBOP (2 equiv.) and DIPEA (10 equiv.) in DMSO for 2 days. Trifluoroacetic acid (70 equiv.) was added to achieve final Boc deprotection in 7 h, giving compound 3 as a TFA salt in 28% yield after purification on preparative reverse phase HPLC (Supplementary Fig. 10).

### Solution NMR spectroscopy

All NMR experiments were performed at 298 K on a Bruker Avance III HD 800 MHz spectrometer equipped with a TCI cryoprobe for enhanced sensitivity. The samples contained 5 µM SOS1-RAS and 0.5 mM aminopiperidine indole ligand with or without 4 molar equivalents of Nb22 (final concentration of 20 µM) in 20 mM potassium phosphate 75 mM NaCl pH 7.0 and 6 % D$_2$O for the lock. The water ligand spectra observed via gradient spectroscopy (water-LOGSY)[33,34] were acquired with the mixing time of 1.5 s, the relaxation delay of 2 s, the acquisition time of 1.28 s and 512 scans. The saturation transfer difference (STD) spectra[35] were acquired with the saturation time of 4 s, the relaxation delay of 5 s, the acquisition time of 1.28 s and 1024 scans. All NMR experiments were recorded, processed, and analyzed in TopSpin 3.6 (Bruker).

### Isothermal titration calorimetry (ITC)

ITC experiments were carried out on a MicroCal *iTC200* system at 25 °C in a buffer containing 25 mM Hepes pH 7.5, 150 mM NaCl and 1% DMSO. The indolo-4-aminopiperidine was resuspended after synthesis at 100 mM in 100% DMSO and diluted to reach the desired concentration in a buffer containing 1% DMSO final. The sample cell was

filled with 30 µM of SOS1 in the presence or absence of 50 µM Nb22. The syringe was loaded with 300 µM of ligand. Titrations comprised 25 × 2 µL injections of ligand into the protein with 90 sec intervals. The raw ITC data were fitted to a single binding site model using the Microcal LLC ITC200 Origin software provided by the manufacturer.

## Reporting summary

Further information on research design is available in the Nature Portfolio Reporting Summary linked to this article.

## Data availability

Crystal structures generated in this study have been deposited in the Protein Databank (PDB) under the PDB accession codes 8BE2 (SOS1-Nanobody77), 8BE3 (KRasG12V-Nanobody84), 8BE4 (SOS1-KRasG12V-Nanobody14) and 8BE5 (SOS1-KRasG12V-Nanobody22-Nanobody75). Source data are provided with this paper.

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

## Acknowledgements

This work was partially supported by Stichting tegen Kanker through project grant number STK 2018-084 granted to A.W. and A.S. We thank INSTRUCT-ERIC and the Research Foundation - Flanders (FWO) for their support to the Nanobody discovery. We thank the staff of the synchrotron SOLEIL (Saint-Aubin, France) for assistance and support in using beamlines PX1 and the staff of the synchrotron DIAMOND (Oxfordshire, UK) for assistance and support in using beamlines IO3 and I24. We thank Océane Rey from the University of Bordeaux, France, for assistance in the preparation of the manuscript.

## Author contributions

Project conceptualization was carried out by B.F., E.P., A.W. and J.S. Construct generation was performed by T.U. Protein production, labeling, and cross-linking were performed by B.F. and E.B. Nanobody selection, BLI, and production were performed by B.F., T.U. and A.W. Crystallography and structural analysis were performed by B.F. and A.W. Kinetic experiments were performed by B.F. and overseen by W.V. and J.S. Cell-based experiments were carried out by A.S. and A.A.S. Synthesis of small-molecules were carried out by S.G. and S.B. NMR measurements were performed by A.N.V. Manuscript writing was undertaken by B.F., A.N.V., T.Z., V.K., S.B., W.V., A.A.S., E.P., A.W. and J.S.

## Competing interests

VIB has applied for a patent, covering the CHiLL & DiSCO technology (application number WO2016/012363A1; Inventors A.W., S.T., J.S.). The remaining authors declare no competing interests.
