## [Peer Review File · Nature Communications]

Allosteric nanobodies to study the interactions between SOS1 and RASREVIEWER COMMENTS

Reviewer #1 (Remarks to the Author):

The authors describe how they developed allosteric nanobodies to either stabilize or disrupt protein-protein interactions (PPIs) in the SOS1•RAS complex. They used a method called ChILL (immunization of cross-linked SOS1•RAS complex), a method previously reported to isolate PPI modulating nanobodies by the author group, to isolate connective and allosteric nanobody stabilizers of SOS1•RAS complex by binding to the complex conformational epitopes exposed on the complex. They then generated the immunized libraries using yeast surface display system to isolate nanobodies that specifically bind to SOS1 (Nb77), RAS (Nb84), or SOS1•RAS complex (Nb14, Nb22). They determined the tertiary structures of the nanobody-the respective antigen complexes to understand the mechanism of action how each nanobody affects the SOS1•RAS PPI interactions and the nucleotide exchanges rates. Overall, the results are generally convincing and support the utility of PPI-modulating nanobodies to gain insight into the nucleotide exchange rates in vitro.

However, this study lacks novelty and significance in general for publication in Nature communications. There are previous reports of developing single-domain antibodies or non-immunoglobulin binders that block the interactions of oncogenic RAS mutants with effector proteins (e.g., Nature Communications (2018), 9:3169) and nucleotide exchange factors like SOS1 (e.g., Nature Communications (2019), 10: 2607). Additionally, there are some SOS inhibitors in clinical trials. Even though this study provides allosteric nanobodies that modulate the PPIs of SOS1•RAS complex by different kinetic mechanisms, the magnitude of such modulation is very marginal, resulting in only minor effects on the RAS downstream signaling (Fig. 4). Most critically, there are no in vitro and in vivo antitumor results of the isolated nanobodies and comparison experiments in antitumor efficacy between the nanobodies and already known SOS1 inhibitors and/or RAS mutant inhibitors. Therefore, the allosteric nanobodies may need to be further engineered in terms of affinity and PPI epitope validation to show some meaningful pharmacological effects in vitro and in vivo.

Reviewer #2 (Remarks to the Author):

The authors describe the ChILL and DisCO methods for discovering nanobodies that can interact with single protomers or complexes. They provide structural and functional characterization of four different nanobodies targeting RAS, SOS1, or the SOS1:RAS complex.

While this manuscript contains interesting data, it is premature for publication, and the following concerns should be addressed:

The methodologies for generating nanobodies are well-established, and the novelty here is the use of chemical crosslinkers to facilitate the discovery of nanobodies that interact with complex interfaces. The authors should acknowledge other efforts that use crosslinking to stabilize proteins targeted by nanobody discovery, such as (<https://pubmed.ncbi.nlm.nih.gov/36095215/>).

In the introduction, the authors state that "adequate tools are missing for in vitro and in vivo characterization and validation" of protein-protein interactions. However, there are many available tools for studying PPIs in vitro and in vivo, such as complementation, NanoBiT, NanoBRET, FRET, alphaeliza, ELISA, ITC, and SPR. The statement is difficult to rationalize and should be revised.

Figure 1 is overly simplistic, and data from the extended data should be incorporated.

In Extended Data Figure 2, it is clear that glutaraldehyde causes new bands to form, but it is not clear if these bands represent complexes similar to the native complexes. Additional characterization is needed, such as gel filtration to compare retention time with the native complex, and intact mass spectrometry to confirm the mass of the covalent particle.

In Figure 4, the hypothesis being tested is that addition of NB77 or NB84 decreases pERK. Statistical comparison of GFP (control) versus Nb should be done to prove or disprove the null hypothesis.

In Figure 5, the overall claim is that Nb22 increases the affinity of an aminopiperidine. The data in support of this claim are weak. While the Water-LOGSY experiment shows enhancement of the interaction, the change in affinity (Kd) is not clear from what is shown, and titration experiments should be done. For the IC50 comparison, it is not clear that there is a statistical difference between the two curves, and 95% confidence intervals should be reported.

Several grammatical errors were found in the writing and should be corrected.

Overall, this manuscript has potential but requires further work before publication.

Please find our answers to all reviewers comments between the lines below (highlighted in green)

REVIEWER COMMENTS

Reviewer #1 (Remarks to the Author):

The authors describe how they developed allosteric nanobodies to either stabilize or disrupt protein-protein interactions (PPIs) in the SOS1•RAS complex. They used a method called ChILL (immunization of cross-linked SOS1•RAS complex), a method previously reported to isolate PPI modulating nanobodies by the author group, to isolate connective and allosteric nanobody stabilizers of SOS1•RAS complex by binding to the complex conformational epitopes exposed on the complex. They then generated the immunized libraries using yeast surface display system to isolate nanobodies that specifically bind to SOS1 (Nb77), RAS (Nb84), or SOS1•RAS complex (Nb14, Nb22). They determined the tertiary structures of the nanobody-the respective antigen complexes to understand the mechanism of action how each nanobody affects the SOS1•RAS PPI interactions and the nucleotide exchanges rates. Overall, the results are generally convincing and support the utility of PPI-modulating nanobodies to gain insight into the nucleotide exchange rates in vitro.

However, this study lacks novelty and significance in general for publication in Nature communications. There are previous reports of developing single-domain antibodies or non-immunoglobulin binders that block the interactions of oncogenic RAS mutants with effector proteins (e.g., Nature Communications (2018), 9:3169) and nucleotide exchange factors like SOS1 (e.g., Nature Communications (2019), 10: 2607). Additionally, there are some SOS inhibitors in clinical trials. Even though this study provides allosteric nanobodies that modulate the PPIs of SOS1•RAS complex by different kinetic mechanisms, the magnitude of such modulation is very marginal, resulting in only minor effects on the RAS downstream signaling (Fig. 4). Most critically, there are no in vitro and in vivo antitumor results of the isolated nanobodies and comparison experiments in antitumor efficacy between the nanobodies and already known SOS1 inhibitors and/or RAS mutant inhibitors. Therefore, the allosteric nanobodies may need to be further engineered in terms of affinity and PPI epitope validation to show some meaningful pharmacological effects in vitro and in vivo.

Lack of novelty: Reviewer #1 refers to previous reports on single-domain antibodies and non-immunoglobulin binders that have been selected against oncogenic RAS mutants or particular effector proteins to identify inhibitors that **block their interactions**. This manuscript has a completely different focus on new technologies and on the discovery of (allosteric) Nanobodies that do not disrupt but rather bind and stabilize protein-protein interactions (PPIs) taking the transient SOS1•RAS binary complex as a proof-of-concept example. And both the technology and the results we obtained on the SOS1•RAS complex are novel. As a matter of fact, our lab pioneered the use of allosteric Nanobodies to stabilize and modulate PPIs. Ground-breaking, we developed the first antibody (Nb35) that stabilizes the interaction between a GPCR and a G protein to form a stable transmembrane signalling complex and this structure made it to the Nobel floor^{1,2}. We later also used Nbs to stabilize the interaction between PINK1 and ubiquitin, a transient enzyme•substrate

complex¹. This manuscript under review for the first time reports in a systematic way how Chill & Disco technology can be used to induce, select and separate Nbs that bind, stabilize and modulate the properties of a transient protein interaction, the SOS1•RAS complex. In conclusion, the Nanobodies described in this manuscript are not ‘me-too’ antibodies that simply copy properties that have been described before, i.e inhibition by steric hindrance. Rather, we introduce a new approach to study the interactions between small G proteins and the regulatory proteins they bind by allosteric modulation.

The magnitude of the allosteric modulation is very marginal. For two of our Nanobodies, we measured allosteric agonism of the nucleotide exchange rate, resulting in a 10 to 20 fold efficacy-increase. Reviewer #1 considers this very marginal but he/she gives no frame of reference. This is particularly unfair in this case because in this same manuscript we (for the first time) provide a genuine estimate of the dynamic range of the allostery in the SOS1•RAS system. Indeed, we included the natural allosteric modulator of the SOS1•RAS interaction -a second GTP bound KRas molecule- as an internal control in our kinetic studies and observed a similar allosteric effect (Fig 3), indicating that our Nbs act within the same dynamic range of allosteric modulation imprinted in this system by nature. In fact, one of the key conclusions of our manuscript is that our allosteric Nbs increase SOS1-catalyzed nucleotide exchange to a similar extent than the natural allosteric modulator.

There are no *in vitro* and *in vivo* experiments to judge antitumor efficacy. Our work is a mechanistic enzymology study focussing on novel methods to study the allosteric interactions between small G proteins and their regulatory proteins. We have chosen SOS1•KRas as a proof-of-concept system because of its relevance in the cancer field but **it has never been our intention to make anti-cancer drugs**. Moreover, KRas is an intracellular target, and it will be extremely difficult to get our Nbs inside cells to treat tumours. The good reason to include cellular data was to demonstrate that Nanobodies can conveniently be expressed as intrabodies inside cells to translate observations made in the test tube to experiments inside living cells. However, it has never been our intention to collect pre-clinical data. To avoid further misunderstandings on this point, we rephrased the title: ‘Using competitive, connective, and allosteric Nanobodies to study the interactions between SOS1 and RAS’

Reviewer #2 (Remarks to the Author):

The authors describe the ChILL and DisCO methods for discovering nanobodies that can interact with single protomers or complexes. They provide structural and functional characterization of four different nanobodies targeting RAS, SOS1, or the SOS1:RAS complex.

While this manuscript contains interesting data, it is premature for publication, and the following concerns should be addressed:

The methodologies for generating nanobodies are well-established, and the novelty here is the use of chemical crosslinkers to facilitate the discovery of nanobodies that interact with complex interfaces. The authors should acknowledge other efforts that use crosslinking to stabilize proteins targeted by nanobody discovery, such as (<https://pubmed.ncbi.nlm.nih.gov/36095215/>).

We have carefully read the Hart *et al.* (2022) manuscript and we noticed that these authors use chemical cross linking to obtain cross-link maps for PI3K α in complex with different nanobodies to be analyzed by mass spectroscopy. We used cross-linking to prepare immunogens for immunizations. It thus appears to us that the Wang lab and our lab used chemical cross-linking for completely different

and unrelated technical and scientific purposes. Therefore, we don't see the relevance of referring to the Hart paper. (<https://pubmed.ncbi.nlm.nih.gov/36095215/>)

In the introduction, the authors state that "adequate tools are missing for in vitro and in vivo characterization and validation" of protein-protein interactions. However, there are many available tools for studying PPIs in vitro and in vivo, such as complementation, NanoBiT, NanoBRET, FRET, alphaeliza, ELISA, ITC, and SPR. The statement is difficult to rationalize and should be revised.

We fully agree with referee 2 on this point and reformulated our statement accordingly: 'Several methods have been developed for studying PPIs in vitro and in vivo, such as complementation, NanoBiT, NanoBRET, FRET, Alphasisa, ELISA, ITC, and SPR but adequate tools are missing to modulate these interactions.'

Figure 1 is overly simplistic, and data from the extended data should be incorporated.

We agree with referee 2 that Figure 1 gives a simplified overview of the process. This corresponds to our aim to present a schematic figure that clearly explains the principles of DISCO for a non-specialist audience. Accordingly, we rephrased the title of this figure: 'Rationale of DisCO for partitioning target-specific Nbs that selectively bind to the SOS1-RAS complex from those that only bind to the separate SOS1 or RAS protomers'. In the revised legend, we also refer to extended figure 3 that provides more (practical) details on the DISCO method. If the editor wishes, we can include the extended figure 3 as an extra panel to Figure 1.

In Extended Data Figure 2, it is clear that glutaraldehyde causes new bands to form, but it is not clear if these bands represent complexes similar to the native complexes. Additional characterization is needed, such as gel filtration to compare retention time with the native complex, and intact mass spectrometry to confirm the mass of the covalent particle.

The aim of our CHILL and DISCO platform is to raise diverse Nanobodies with different binding properties towards the PPI: connective and allosteric stabilizers but also competitive inhibitors. Accordingly, we (by purpose) immunize with heterogeneous samples that contain the cross-linked complexes (to trigger allosteric PPI binders), but also the non-crosslinked protomers (for triggering competitive inhibitors). Similar experiments that we performed previously to obtain allosteric antibodies that stabilize other PPIs^{1,2} confirm that such heterogeneous samples contain antigen in an association that is very similar to the native complex. Moreover, heterogeneous samples also leave room for serendipitous discoveries on PPIs of non-anticipated composition or stoichiometry. Finally, the selection process occurs with non-crosslinked proteins, which guarantees that only Nbs binding to "relevant native conformations" are selected. As a matter of fact, the detailed kinetic analysis presented in this study (Fig 3) also demonstrates that the Nanobodies that we elicit by immunization with these heterogeneous samples behave very similar compared to the natural allosteric modulator of the SOS1-RAS complex.

However and in line with the suggestion of referee 2, we routinely use western blots, developed with separate antibodies that recognize the different protomers to confirm that the associating proteins migrate as a complex of the expected MW on SDS-page.

In Figure 4, the hypothesis being tested is that addition of NB77 or NB84 decreases pERK. Statistical comparison of GFP (control) versus Nb should be done to prove or disprove the null hypothesis.

We fully agree with referee 2 that a statistical comparison was missing and we updated Figure 4 accordingly.

In Figure 5, the overall claim is that Nb22 increases the affinity of an aminopiperidine. The data in support of this claim are weak. While the Water-LOGSY experiment shows enhancement of the interaction, the change in affinity (K_d) is not clear from what is shown, and titration experiments should be done.

Given that the Water-LOGSY signal not only depends on the amount of the ligand bound, but also on the total ligand concentration and the exchange rate between the free and protein-bound ligand forms, this experiment is not suitable for the K_d determination, as the signal changes during the titration will depend on a non-trivial combination of these three parameters. However, comparison of the peak intensities at the same ligand concentration serves as a good proxy for the relative changes in the binding affinity, and this method is routinely used in the ligand-based NMR screening to rank primary hits (e.g. Lepre et al. *Chem. Rev.* 2004, 104, 3641-3675). Thus, while we cannot put numbers on the K_d values from Water-LOGSY experiments, we can confidently conclude that Nb22 indeed enhances the ligand binding because we use the same ligand in these comparisons. The sheer fact that we observe a binding signal in both STD and Water-LOGSY means that the ligand binding is weak (K_D in the 10 μM – 1 mM range). The “only” 2-fold increase in the Nb22-mediated binding affinity might not sound like a lot, but this is a great deal in a typical NMR fragment screening project, because this difference is significant and it makes the difference between detecting a hit or missing it altogether.

To reassure referee 2, we analyzed the impact of Nb22 on the binding constant for aminopiperidine by also performing ITC experiments (Figure 5b). Consistent with all our results obtained prior to revision, we measured a threefold increase in affinity of the complex for aminopiperidine upon binding of Nb22 (K_d improves from 10.7 μM to 3.3 μM).

For the IC50 comparison, it is not clear that there is a statistical difference between the two curves, and 95% confidence intervals should be reported.

As asked by referee 2, we included a statistical analysis in the legend of Fig 5.

Several grammatical errors were found in the writing and should be corrected.

We asked a native speaker to read the text and correct the errors.

1. Schubert, A.F. et al. (2017) Structure of PINK1 in complex with its substrate ubiquitin. *Nature* **552**, 51-56.
2. Rasmussen, S.G. et al. (2011) Crystal structure of the b_2 adrenergic receptor-Gs protein complex. *Nature* **477**, 549-555.

REVIEWERS' COMMENTS

Reviewer #1 (Remarks to the Author):

The authors addressed my comments properly.

Reviewer #2 (Remarks to the Author):

Most of my comments were adequately addressed. Figure 1 was not changed and this reviewer still has the opinion that it lacks sophistication for a Nature Communications manuscript. I defer to the editor on how to proceed with this.